# Efficacy of fluopyram applied by chemigation on controlling eggplant root-knot nematodes (*Meloidogyne* spp.) and its effects on soil properties

**Jinzhao Li, Cancan Wang, Saqib Hussain Bangash, Haiou Lin, Dongqiang Zeng, Wenwei Tang**  *

Guangxi Key Laboratory of Agric-Environment and Agric-Product Safety, Agricultural College, Guangxi University, Nanning, Guangxi, People's Republic of China

* wenweitg@163.com

**Data Availability Statement:** All relevant data are within the manuscript.

**Funding:** Dongqiang Zeng received the award supported by the Key Research and Development

## Abstract

The root-knot nematode (*Meloidogyne* spp.) is one of the major challenges in eggplant (*Solanum melongena* L.) production. Fluopyram, known to be an effective fungicide, is also used for controlling root-knot nematode. However, in China, little information is currently available regarding the efficacy of fluopyram via chemigation against root-knot nematode and its effects on soil properties. For this, the objective of this work was to test mortality of root-knot nematode, functional diversity of soil microbial community, activity of soil enzyme after fluopyram applicated by chemigation. The results of two field experiments revealed that concentration of 60 g·ha$^{-1}$ fluopyram applied with 200 L·ha$^{-1}$ irrigation water at 2 L·h$^{-1}$ flow velocity was the most effective chemigation parameters for controlling eggplant against root-knot nematode. The functional diversity of the soil microbial community was significantly affected by fluopyram. The activities of soil urease and $\beta$—glucosidase decreased during the initial stages but recovered at later stages. In brief, fluopyram has advantageous for the efficient control of root-knot nematode with no deleterious effects on soil properties as well as chemigation is positive for application in karst landscape in Guangxi.

## Introduction

The root-knot nematode (RKN, *Meloidogyne* spp.) is one of the best known and the most harmful plant parasitic nematodes that causes serious damage to important agricultural crops, particularly eggplant (*Solanum melongena*) [1,2]. RKN penetrates growing root tips and forms multinucleate giant cells in damaged tissues, leading to gall formation, resulting in forked and defective eggplants [3], and subsequently disrupting physiological processes [4]. The damage caused by RKN is more frequent during hot climatic conditions and results in massive losses in net productivity [5].

Fluopyram (*N*-[2-[3-chloro-5-(trifluoromethyl)-2-pyridyl]ethyl]-α, α, α—trifluoro-ortho-toluamide) was initially developed as a fungicide by Bayer Crop Science in 2012 and was

Program of Guangxi (Guike AB16380118). The funders had no role in study design, data collection and analysis, decision to publish, or preparation of the manuscript.

**Competing interests:** The authors have declared that no competing interests exist.

mainly used to control grey mould and powdery mildew in grapes but was also used against fungi in many other fruits and crops [6–8]. Recently, some researchers have reported that fluopyram contains a succinate dehydrogenase inhibitor (SDHI), which can be useful in nematode control [9,10]. Fluopyram is registered in China as nematicide for use in tomato by soil drenching. Although fluopyram is widely used to control nematode reproduction, the RKN control efficacy via chemigation has been rarely reported.

Guangxi is a part of southwest karst region in China which area is about 5.5 million km$^2$, accounting for 15.97% of national karst area [11]. Karst area is dreadful for agriculture development with thin soil layer, faint ground water impact and serious water leaking [12]. To overcome the demerit, irrigation systems have been used successfully for vegetable production over many years. The sown area (SA) of major farm crops in Guangxi was 6.15 million ha in 2016, and the water-saving irrigation area (WSIA) was 1.03 million ha. Within this WSIA, the sprinkling-drip irrigation area (SDIA) was 0.1 million ha, the ratio of WSIA / SA was 16.77%, which increased by 8.12% from last year, and the ratio of SDIA / IA was 9.85%, which increased by 39.91% from last year [13].

In this study, 'chemigation' consists of installing a chemical bucket to the original drip irrigation system and can be a conveniently applied method because drip irrigation is widely used in farms in Guangxi. Initially, such irrigation systems were used as a water-saving technique, whilst they are currently used for fertilizer and insecticide applications worldwide [14,15]. The benefits of chemigation are various, and this method has reduced insect pest problems more than traditional foliar applications or other methods. It is reasonable to assume that fluopyram applied by chemigation is efficacious for RKN control. Many studies have examined the risks of pesticides to soil organisms. Fluopyram was first developed as a fungicide, and it was confirmed to change soil microbial communities [16]. Furthermore, changes in the soil environment caused by fungicides usually lead to reductions in the abundance and diversity of microorganisms [17]. During the cycling of nutrients, some hydrolytic enzymes are involved ($\beta$-glucosidase, urease, and phosphatase linked to C, N, and P, respectively). These enzymes are sensitive indicators of changes in the soil properties and show a strong relationship with the content and quality of soil organic mulches. It is reasonable to assume that fluopyram affects soil health and productivity.

Therefore, the aim of the study was to enhance fluopyram efficacy against eggplant RKN by chemigation and to investigate the effects of fluopyram on the development of eggplant roots, the functional diversity of the soil microbial community, the activity of soil enzymes and the terminal residues of fluopyram in eggplant fruit to verify its safety for both eggplants and soil ecosystems. The study will also aid in promoting fluopyram for control of eggplant RKN by application via chemigation.

## Materials and methods

### Instruments and reagents

The chemigation system was based on the available drip irrigation system (Jiejiarun Agriculture Technology Company, Nanning, Guangxi, China) in the experimental field. Residues of fluopyram in eggplant fruit were analysed by gas chromatography (GC), using a Spherisorb DB-17 column (Agilent). Detection was performed with an ECD detector using the fluopyram standard (purity was 99.4%, Ehrenstorfer GmbH Co.). A 41.7% fluopyram suspension concentrate (SC) was used in the field experiment (Bayer Crop Science Co., Ltd.), and the other reagents were of analytical grade (Guangzhou Chemical Reagents Factory Co., China).

## Site description

Field experiments were carried out at Jinling Village (22.92˚ N, 108.05˚ E) in Nanning, Guangxi, China, where a hot climate exists with an average temperature of 22.4˚C. In May, the precipitation was 29.4 mm, and the total illumination was 109.6 h; in August, the average temperature was 28.0˚C, the total precipitation was 124.9 mm and the total illumination was 171.4 h. The climatic characteristics of the experimental area were suitable for propagation of root-knot nematodes. The total planting area was approximately 4.2 ha. The soil type was loamy with a soil pH of 7.8. Fertilizer applications and other agronomic practices were carried out regularly as needed.

## Experimental design and treatment application

The field experiment was conducted in commercial fields in May and August 2018. The whole field was divided into two equal portions for separate experiments during May and August. Each experimental plot was 300 m², which was divided into 3 random treatment plots, and the experimental plots were separated from each other by a 70 m² protective plot (Fig 1A).

In the field, a drip tape of chemigation was watered one adjacent row of eggplant in the boll stage and was covered with black, virtually impermeable film. Eggplants were separated from each other by 0.6 m. The chemigation system was based on the available drip irrigation system with an added bucket for applying fluopyram (Fig 1B). The principle of operation was siphonage. After germination of the 4th true leaf, 41.7% fluopyram SC was applied only once in low,

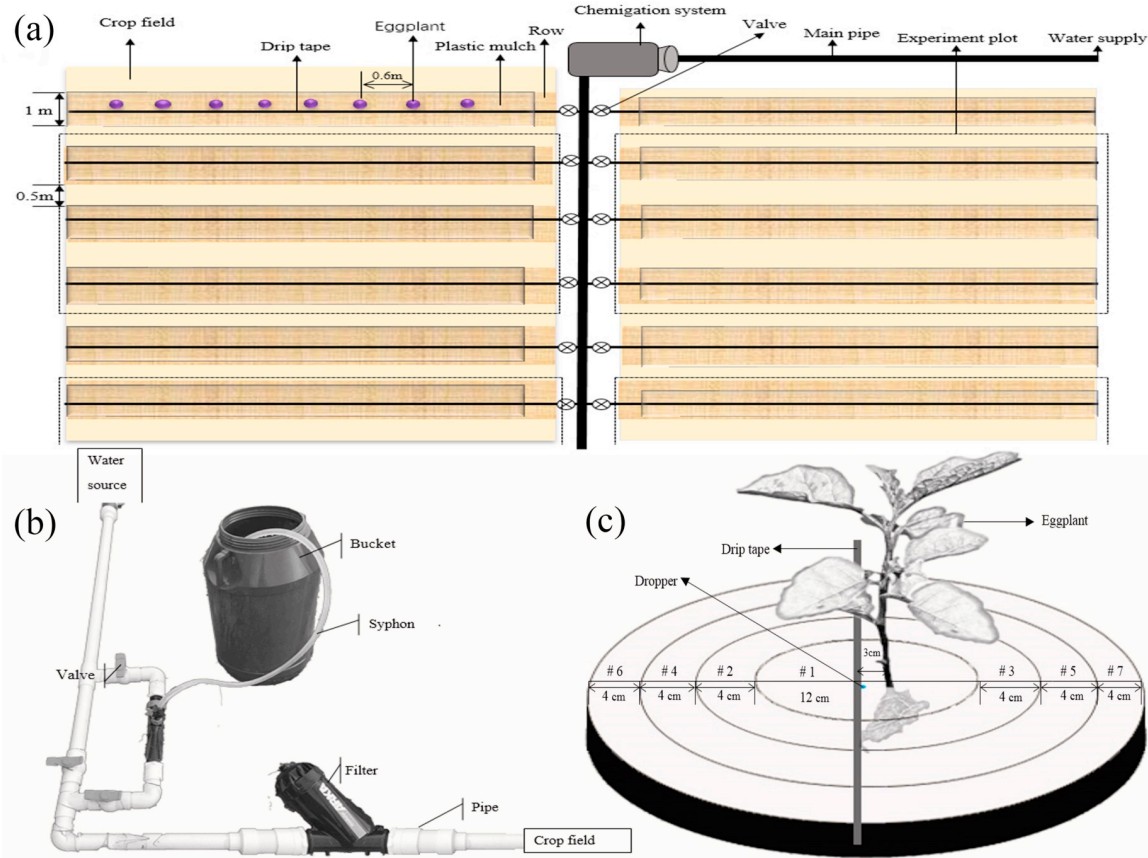

**Fig 1. Survey of field experiment.** (a) Survey of field experiment plot. (b). Detail of chemigation system. (c) Sampling spots.

moderate and high doses (40, 60 and 80 g·ha$^{-1}$, LD, MD and HD, respectively) with low and high irrigation water volumes (100 and 200 L·ha$^{-1}$, LIWV and HIWV, respectively) at low and high flow velocities (1 and 2 L·h$^{-1}$, LFV and HFV, respectively) per the appropriate procedures [18]. Fluopyram was applied with 75 and 150 L·ha$^{-1}$ irrigation water for each water volume treatment, then was applied with 25 and 50 L·ha$^{-1}$ irrigation water to rinse the tape. The drip tape was cut at the end and closed using end caps.

The sampling area was divided into 7 spots according to the distance from the spot to the dripper (Fig 1C). Then, spot 1 (a circular area with a radius of 6 cm from the dripper, centre area, CA); spots 2 and 3 (torus-shaped areas with distances to dripper of 6 cm and widths of 4 cm, close-distance area, CDA); spots 4 and 5 (torus-shaped areas with distances to dripper of 10 cm and widths of 4 cm, mid-distance area, MDA); and spots 6 and 7 (torus-shaped areas with distances to the dripper of 14 cm and widths of 4 cm, long-distance area, LDA) were established.

## Control efficacy for RKN

On each sampling date (7, 15 and 30 days after treatment, DAT), 500 g soil samples were collected consisting of soil samples from 7 sampling spots around a plant. The separation method for RKN used a Baermann funnel, described by [19]. Soil samples from each spot were separated and replicated 3 times. RKN populations were counted by microscope. RKNs were considered to be dead if they did not respond to being touched with a small probe. The RKN efficacy can be described by the following equations based on [20]:

$$\text{Corrected mortality (\%): } m = (m_t - m_c) / (1 - m_c) \times 100 \tag{1}$$

$$\text{General mortality(\%): } M = \sum_{i=1}^{4} m_i \cdot si/S \tag{2}$$

where $m_t$ is nematode mortality (%) from fluopyram treatment, and $m_c$ is the blank control mortality (%); $m_i$ is nematode mortality (%) from different sampling areas; $s_i$ is the area (cm$^2$) of different sampling areas, $i$ = 1,2,3 and 4, representing the centre, close-distance, mid-distance and long-distance areas; and $S$ is the area (cm$^2$) of the entire sampled area.

## Functional diversity of soil microbial community determination

BIOLOG Eco plates (BIOLOG Inc., Hayward, USA) were used to measure the soil microbial physiological profiles and functional diversity of the microbial community [21]. They contain three replicate sets of 31 carbon substrates which are degradable by different soil microbial. The following microbial indices were calculated for each plate and sample: AWCD (Average Well Color Development, overall microbial metabolic capacity), Shannon index ($H'$, substrate richness), Simpson index ($D$, functional diversity index), and McIntosh index ($U$, index of evenness) [22].On each sampling date (3, 7, 14, 21 and 30 DAT), 500 g soil samples were randomly collected for each treatment. The AWCD, Shannon index, Simpson's diversity index and the McIntosh index were determined by calculating the mean of the absorbance value for every well after 96 h incubation, which corresponded to the time of maximal microbial growth in the BIOLOG Eco plates as determined by a BIO-TEKElx 808 automated micro plate reader (BIOLOG Inc., Hayward, USA) [17].

$$\text{AWCD} = \sum OD_1 / 31 \tag{3}$$

$$\text{Shannon index: } H' = - \sum P_i \times \ln(P_i) \tag{4}$$

$$\text{Simpson index: } D = \sum \left[ n_i(n_i - 1) \,/ N(N - 1) \right] \tag{5}$$

$$\text{McIntosh index: } U = \sqrt{\sum (n_i)^2} \tag{6}$$

where $OD_i$ is the optical density value from each well after subtracting the value of the blank (water). $pi$ is the ratio of microbial activity on each substrate ($OD_i$) to the sum of the microbial activities on all substrates, $\Sigma OD_i$. $n_i$ is the absorbance value, $N$ is the total absorbance value for all wells, and the Simpson index is expressed as the reciprocal (1/D).

## Soil enzyme activities determination

The activities of three soil enzymes (urease, $\beta$-glucosidase and alkaline phosphatase) were determined to evaluate the ecotoxicology of fluopyram. On each sampling date (3, 7, 14, 21 and 30 DAT), 500 g soil samples were randomly collected for each treatment and were screened for soil enzyme analysis. The operation followed the instructions of a soil enzyme assay kit (SOLABIO, CO). Finally, the mixtures were measured with a spectrophotometer (UV-2600, Shimadzu, Japan) at 400 nm (soil $\beta$-glucosidase), 630 nm (soil urease) and 660 nm (soil alkaline phosphatase).

$$\text{Inhibition rate: } p_i = (1 - a_i/a_c) \times 100\% \tag{7}$$

where $p_i$ is the inhibition rate of fluopyram on soil enzyme activity; $a_i$ is the activity of a soil enzyme after treatment from different soil sampling spots, from spot 1 to 7; and $a_c$ is the activity of a soil enzyme in the control.

## Fluopyram residues in eggplant fruit

The determination method was based on [23] with some modifications The eggplant fruit samples were collected and crushed at 30 DAT, and a 10.0 g sample was weighed and placed in a centrifugal tube. A volume of 20.0 mL acetonitrile with 4.0 g NaCL was added and mixed for 1 minute. After 30 minutes, the samples were centrifuged at 1000 xg for 5 minutes. The supernatant was collected and dried by rotary evaporation at 70°C, then 2.0 mL n-hexane was added and covered. A Florisil SPE Column was leached with 3.0 mL leachate (n-hexane: acetone = 7:3, v/v), and then 3.0 mL n-hexane and the sample solution were added. A 10 μl sample was injected into the GC system for measurement.

## Statistical analysis

All quantitative data were presented as the mean ± SE of at least three independent experiments by Tukey's test to determine the differences using SPSS 20.0. A *P*-value of 0.05 was considered to be statistically significant.

## Results

### Exploration of the effect factor for the corrected mortality of RKN

The corrected RKN mortalities resulting from different treatments are presented in Table 1. The results from the field experiments showed that the corrected RKN mortalities for different sampling areas were related to the distances from drippers. The general RKN mortalities are presented in Table 2. The corrected mortalities in the LD group were significantly lower than those in other dose groups for each sampling day. There was a significant difference between the corrected mortalities in the HIWV group and those in LIWV group at the same fluopyram

**Table 1. Corrected mortalities from fluopyram applied by chemigation to control RKN in the field.**

| DF | IWV | FV | Days after treatment | | | | | | | | | | | |
|---|---|---|---|---|---|---|---|---|---|---|---|---|---|---|
| | | | 7 | | | | 15 | | | | 30 | | | |
| | | | CA | CDA | MDA | LDA | CA | CDA | MDA | LDA | CA | CDA | MDA | LDA |
| 40 | 100 | 1 | 43.42 ±2.22c | 30.74 ±2.59c | 20.66 ±3.47d | 14.04 ±4.15g | 51.57 ±2.46d | 37.1±1.88h | 28.24 ±1.99d | 19.68 ±4.89d | 62.56 ±2.72c | 51.18 ±2.13d | 36.85 ±1.62e | 31.75 ±2.32d |
| | | 2 | 46.03 ±2.22c | 31.72 ±2.38c | 20.99 ±4.23d | 14.49 ±4.73g | 52.17 ±1.77d | 38.88 ±2.04fg | 29.43 ±2.14d | 20.87 ±4.04d | 63.03 ±1.82c | 52.13 ±1.58d | 38.15 ±2.63e | 29.72 ±2.51cd |
| | 200 | 1 | 43.42 ±2.21c | 34.65 ±2.64c | 24.24 ±2.45d | 16.76 ±4.45g | 52.76 ±3.47d | 44.19 ±4.13efg | 36.51 ±3.12c | 23.52 ±3.21d | 62.56 ±3.03c | 58.29 ±2.09c | 41.94 ±3.03de | 34.83 ±2.43cd |
| | | 2 | 43.42 ±2.09c | 35.30 ±2.35c | 25.87 ±3.40d | 19.36 ±2.78fg | 52.17 ±0.51d | 45.37 ±2.23efg | 38.58 ±2.60c | 27.07 ±4.57cd | 63.03 ±2.74c | 59.00 ±2.53c | 44.05 ±3.38de | 36.97 ±4.11cd |
| 60 | 100 | 1 | 59.68 ±3.09b | 44.73 ±3.39b | 34.97 ±2.99c | 24.79 ±1.81ef | 65.75 ±2.26bc | 50.69 ±1.83def | 39.76 ±1.71c | 32.97 ±2.27bc | 72.92 ±0.91b | 56.62 ±3.34cd | 47.39 ±3.20d | 38.63±4.08 bc |
| | | 2 | 60.33 ±2.68b | 45.38 ±3.03b | 35.95 ±3.54c | 27.17 ±3.75de | 66.34 ±4.02bc | 52.46 ±4.57de | 41.53 ±4.23c | 32.97 ±3.51bc | 73.41 ±3.24b | 63.98 ±2.28bc | 47.39 ±4.05d | 39.81 ±3.03bc |
| | 200 | 1 | 59.68 ±3.71b | 48.95 ±3.15b | 42.45 ±2.55b | 37.07 ±3.30bc | 65.16 ±3.10c | 58.07 ±3.27bcd | 49.51 ±2.40b | 40.94 ±2.23ab | 73.41 ±2.85b | 67.06 ±2.95b | 55.45 ±4.38c | 44.79 ±2.51b |
| | | 2 | 60.98 ±1.84b | 50.25 ±2.45b | 44.08 ±1.84b | 38.22 ±1.94bc | 66.93 ±3.34bc | 58.36 ±3.21bcd | 49.51 ±1.41b | 43.01 ±4.52a | 73.91 ±2.97b | 68.24 ±2.41b | 56.40 ±2.21c | 44.79 ±2.35b |
| 80 | 100 | 1 | 70.74 ±2.22a | 55.78 ±3.11a | 43.02 ±1.35b | 30.97 ±2.42cde | 75.20 ±2.46ab | 65.45 ±3.28bc | 52.76 ±1.49b | 40.65 ±2.23ab | 82.46 ±1.62a | 73.46 ±1.52a | 63.74 ±3.74b | 51.42 ±2.72a |
| | | 2 | 70.09 ±1.30a | 58.06 ±2.13a | 42.77 ±3.99b | 32.69 ±4.37cd | 76.97 ±3.48a | 65.15 ±3.10abc | 50.98 ±3.72b | 40.94 ±0.57ab | 82.46 ±2.10a | 74.88 ±2.01a | 66.11 ±1.28ab | 52.84 ±2.35a |
| | 200 | 1 | 71.39 ±1.50a | 60.01 ±1.54a | 51.55 ±2.30a | 44.08 ±4.11ab | 76.97 ±1.72a | 68.70 ±3.77ab | 60.43 ±1.97a | 45.96 ±1.83a | 81.99 ±1.97a | 78.20 ±2.32a | 69.91 ±2.55ab | 54.50 ±2.02a |
| | | 2 | 70.09 ±1.68 | 60.66 ±2.14a | 51.55 ±2.30a | 46.68 ±1.20a | 76.38 ±3.47a | 70.18±3.40a | 61.61 ±2.34a | 47.74 ±3.96a | 84.36 ±1.42a | 78.88 ±3.52a | 71.56 ±1.60a | 56.40 ±3.74a |
| | df | | 47 | 95 | 95 | 95 | 47 | 95 | 95 | 95 | 47 | 95 | 95 | 95 |
| | F | | 23.806 | 35.998 | 26.447 | 21.579 | 11.412 | 21.449 | 32.902 | 12.098 | 16.130 | 30.419 | 34.727 | 19.367 |
| | P | | < 0.001 | < 0.001 | < 0.001 | < 0.001 | < 0.001 | < 0.001 | < 0.001 | < 0.001 | < 0.001 | < 0.001 | < 0.001 | < 0.001 |

DF represents the dose of fluopyram (g·ha$^{-1}$), IWV represents irrigation water volume (L·ha$^{-1}$), and FV represents the flow velocity (L·h$^{-1}$). All data represent means ± SE. Values followed by different letters in the same column indicate significant differences (P < 0.05) according to Tukey's test.

dose and for the same flow velocity. The corrected mortalities in the HFV group were greater than those in the LWV group, but the difference was insignificant. In the HIWV and HFV treatment groups, the general mortalities for the 60 g·ha$^{-1}$ fluopyram treatment were 56.55%, 62.60% and 69.51% at 7, 15 and 30 DAT, respectively.

The mortalities for the 80 g·ha$^{-1}$ treatment were 65.60%, 72.18% and 80.48%, respectively, on the mentioned dates. Although fluopyram at the HD level exhibited better control of RKN, it also substantially affected soil community structures (Fig 2).

Analysis of the main effects and interactions showed that the times (*df* = 3; *F* = 2.013; *P* < 0.0001) and irrigation water volumes (*df* = 2; *F* = 1.080; *P* < 0.0001) were significant factors contributing to the control efficacy for RKN except for the fluopyram rate, while a flow velocity (*df* = 2; *F* = 0.495; *P* < 0.0001) did not exhibit a substantial effect for controlling RKN. The higher water volume (200 L·ha$^{-1}$) and higher flow velocity (2 L·h$^{-1}$) were appropriate parameters for chemigation of fluopyram to control the root-knot nematode.

## Control efficacy of fluopyram on eggplant root-knot nematodes

There were no significant differences among the RKN mortalities from different seasons (Fig 3). The corrected RKN mortality was maintained at 60.98% from 7 to 30 DAT with a peak of

**Table 2. General mortalities of fluopyram applied by chemigation to control RKN in the field.**

| DF | IWV | FV | Days after treatment | | |
|---|---|---|---|---|---|
| | | | 7 | 15 | 30 |
| 40 | 100 | 1 | 37.75 ± 1.93d | 45.46 ± 2.00d | 56.64 ± 1.86d |
| | | 2 | 39.80 ± 1.64d | 46.27 ± 1.76d | 57.25 ± 1.27d |
| | 200 | 1 | 38.70 ± 128d | 48.00 ± 2.28d | 58.12 ± 1.97d |
| | | 2 | 39.12 ± 1.58d | 48.28 ± 3.80d | 58.84 ± 2.24d |
| 60 | 100 | 1 | 53.21 ± 2.31c | 59.26 ± 1.74c | 67.00 ± 1.04c |
| | | 2 | 53.97 ± 2.31c | 59.99 ± 2.55c | 67.51 ± 2.23c |
| | 200 | 1 | 55.17 ± 2.83c | 61.11 ± 2.34bc | 68.95 ± 1.60c |
| | | 2 | 56.55 ± 1.33bc | 62.60 ± 2.00abc | 69.51 ± 0.57bc |
| 80 | 100 | 1 | 63.60 ± 1.65ab | 69.47 ± 1.75abc | 77.42 ± 1.24ab |
| | | 2 | 63.46 ± 1.04ab | 71.15 ± 0.80ab | 77.90 ± 1.64a |
| | 200 | 1 | 66.21 ± 1.03a | 72.21 ± 1.29a | 78.39 ± 1.60a |
| | | 2 | 65.60 ± 1.05a | 72.18 ± 2.50a | 80.48 ± 1.13a |
| | df | | 47 | 47 | 47 |
| | F | | 40.611 | 22.812 | 31.004 |
| | P | | < 0.001 | < 0.001 | < 0.001 |

All data represent means ± SE. Values followed by different letters in the same column indicate significant differences ($P < 0.05$) according to Tukey's test.

73.91% at 30 DAT in the centre area in May, while RKN mortality was maintained at 61.20% from 7 to 30 DAT with a peak of 75.35% at 30 DAT in August. The RKN mortalities for different soil locations exhibited a normal distribution tendency, and the most efficient control was observed at the centre area under the dripper and gradually decreased with distance from centre area.

## Effects of fluopyram on the functional diversity of the soil microbial community

There were similar effects among the data from the two experiments that could be combined for analysis. As shown in Fig 4, there were significant differences between treatment and control in the Shannon and McIntosh indexes at 30 DAT. The AWCDs of the treatments were close to the control level during the experimental period except for those of the first treatment

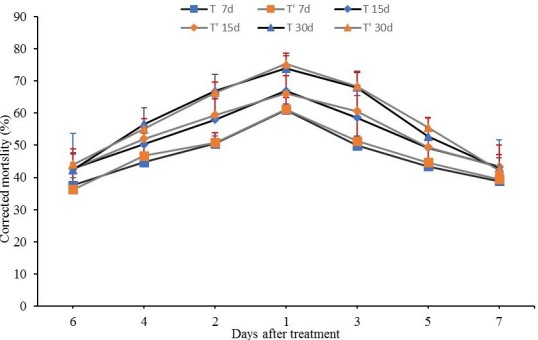

**Fig 2. Effects of different rates of fluopyram on the functional diversity of the soil microbial community.** The dashed line represents the average percentage of control. "*" represents significant differences between two-time treatments and control at each time as measured by Tukey's test (P < 0.05).

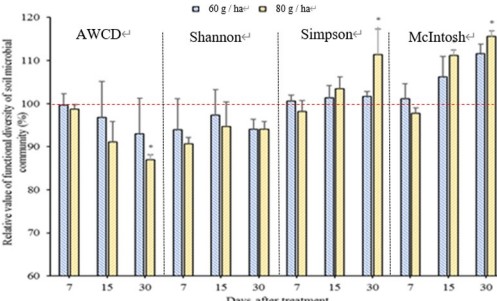

**Fig 3. Corrected RKN mortality of fluopyram applied by chemigation under the most effective parameters on root-knot nematodes in a two-time field experiment.** The blue figure represents the corrected RKN mortality in field experiments in May, 2018; the orange figure represents the corrected RKN mortality in the adjacent experimental field in August, 2018.

at 7 and 30 DAT. The Shannon index of treatments decreased at 30 DAT. The Simpson index for the treatments increased from 7 to 30 DAT, but the differences between the treatments and control were insignificant. The McIntosh index in the treatment groups was significantly higher than in the control group at 30 DAT. As described above, fluopyram changed the soil microbial functional diversity.

## Effects of fluopyram on the activity of soil enzymes

Soil enzymes, especially $\beta$-glucosidase, have a critical role in C mineralization. Similarly, urease and alkaline phosphatase also play critical roles in the N and P cycles, respectively [24]. Therefore, soil enzyme activity could be an indicator of soil biological activity [18,25]. The responses of soil enzyme activities, including urease, $\beta$-glucosidase and alkaline phosphatase, after fluopyram application by chemigation are shown in Fig 5. The activity of soil urease in the treatments showed significant changes relative to the control at 21 DAT, particularly in the centre, close and mid-distance areas (from spot 1 to spot 5) (Fig 5A). The effect of fluopyram on soil urease showed an overall distinct decrease at 7 DAT, while the soil urease activity of the treatments returned to the control level at 30 DAT. Fluopyram significantly inhibited the activity of soil $\beta$-glucosidase in the centre area at 3 DAT (Fig 5B). The effect on the activity of soil $\beta$-glucosidase in the close-distance area increased slightly with the diffusion of fluopyram. The soil $\beta$-glucosidase activity for all areas recovered gradually from 14 to 30 DAT, and no significant

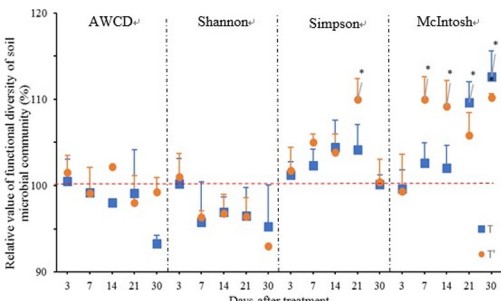

**Fig 4. Variations in the soil microbial community index as affected by fluopyram.** The dashed line shows the average percentage of control. The blue square represents the relative value of treatment on root development in the experimental field in May 2018; The orange circle represents the relative value of treatment on root development in the adjacent experimental field in August, 2018. "*" represents significant differences between treatment and control according to Tukey's test (P < 0.05).

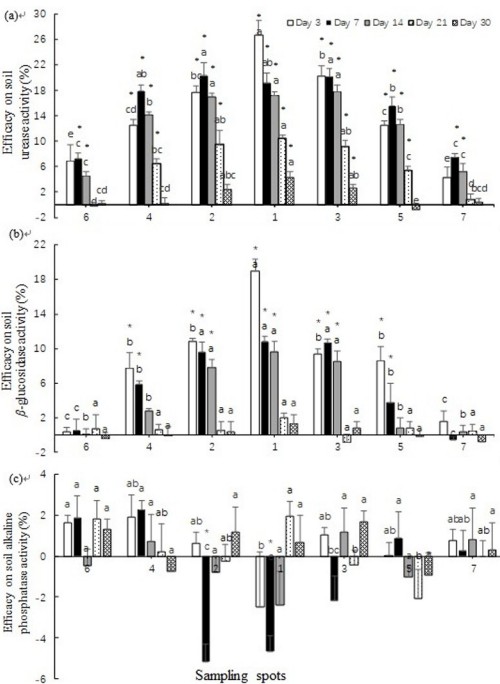

**Fig 5. Effects of fluopyram on soil enzyme activity.** (a) Soil urease. (b) Soil- $\beta$—glucosidase. (c) Soil alkaline phosphatase. Each column is the average value of the triplicates. The standard deviation is illustrated using an error bar. The significant differences among different sampling spots are illustrated using different letters above the columns at the p < 0.05 level via the least significant difference (LSD) test. Significant differences between treatment and control are illustrated using a "*" symbol above the columns at the p < 0.05 level via Tukey's test.

differences were observed between control and treatments at 30 DAT. However, a slight increase in the centre and close-distance areas was observed from 3 to 7 DAT. In this study, the activity of soil alkaline phosphatase seemed to be insensitive to fluopyram (Fig 5C).

## Fluopyram residues in eggplant fruit

Reliable linearity, y = 19612 x -1766.2, was achieved with fluopyram standard dosages in the range from 0.05 to 1.00 ?g/ml with a correlation coefficient ($R^2$) = 0.9878 for fluopyram in all cases. The recovery rates ranged from 80.55% to 84.76%, and the relative standard deviations (RSD) ranged from 3.74% to 10.80%. In all cases, the results from the recovery tests were acceptable and confirmed that the method was sufficiently reliable for fluopyram analysis in this study. The terminal residue (30 DAT) of fluopyram was 0.076 mg·kg$^{-1}$, which was below the maximum residue limit of fluopyram in eggplant fruit, 0.9 mg·kg$^{-1}$.

## Discussion

In previous laboratory test, we have confirmed the control efficacy of fluopyram, compared with major nematicides, such as avermectin, thizaolin and carbosulfan [26]. In this experiment, fluopyram was applied at a dose of 60 g·ha$^{-1}$ with 200 L·ha$^{-1}$ irrigation water and a 2 L·h$^{-1}$ flow velocity that showed substantial control of root-knot nematodes, resulting in 69.51% and 70.22% general mortality at 30 DAT for two continuous field experiments. Appropriate application of nematicide can improve efficiency, reduce the dose and costs. The results were similar in beans when fluopyram was applied at a dose of 91.74 g·ha$^{-1}$ below the seed in furrows, for which the control efficacy was 88.34% for RKN. Likewise, the control efficacy of 10

mg·L$^{-1}$ abamectin SC was 74% on RKN [20]. The RKN mortality from fosthiazate for potato cyst nematodes was 74.85% [27]. Hence, the control efficacy of eggplant RKN for fluopyram applied via chemigation seemed to be acceptable when compared to other popular nematicides. Furthermore, our research indicates that more irrigation water could be instrumental in diffusion area and control efficacy of fluopyram applied by chemigation. This finding is consistent with [28,29]. The volume weight of tested soil was 1.05 g·cm$^{-3}$, and the soil permeability was 0.36 L·h$^{-1}$. When the flow velocity was faster than the soil permeability, chemigation reduced the downward loss of soil moisture and increased the horizontal motion of fluopyram in the irrigation water. This pattern was in accordance with [30,31]. With the popularization irrigation system in Guangxi, these results suggested that fluopyram applied by chemigation systems is highly promising.

BIOLOG ECO plates were used to study the substrate utilization pattern of soil microbial communities [32]. Our investigation demonstrated that fluopyram affected functional diversity of soil microbial community. As described above (Fig 4), AWCD and Shannon index in treatment were decreased while Simpson and McIntosh index in treatment were increased during the incubation period. These findings suggest that fluopyram inhibit the growth of some microbial due to its toxicity and breed dominant population in the soil. Our results are consistent with other researches. fluopyram has a negative impact on microbial respiration, microbial biomass, bacteria (including GP and GN) and fungi [17]. There are two aspects concerned with the major relationship between pesticides and microbial communities in soil. One is that pesticides have a negative impact on the microbial community, affecting the growth and reproduction of microbial [16,33]. The other is that some microorganisms can decompose and use pesticides for their own growth [34].

Activities of soil enzyme activities are considered soil quality/health indicators reflecting changes in biogeochemical cycling and soil organic matter dynamics [35]. In field experiments, the activity of soil urease was inhibited by fluopyram during the early and mid-periods but resumed at later periods (Fig 5A). This dynamic process coincided with the AWCD of the microbial community. Based on this finding, we can suspect that activity of soil urease is associated with microbial abundance. Soil $\beta$-glucosidase is involved in cellulose degradation which is the most abundant polysaccharide in nature [36]. Our research showed that fluopyram has insignificant effect on activity of Soil $\beta$-glucosidase on fruit harvest time (Fig 5B). Similar results have been demonstrated by [37–39]. Soil phosphatase plays a key role in hydrolysing organic phosphate to inorganic form, thereby enhancing the supply of soil phosphorus [40]. In this study, the activity of soil alkaline phosphatase seemed to be insensitive to fluopyram (Fig 5C). A correlation analysis showed that the fluopyram effects on activity of soil enzyme were positively correlated to distance from the irrigation drippers. According to this discovery, eggplant should be planted in the close-distance area where acceptable control efficacy on the root-knot nematode is obtained and fewer negative effects on soil enzymes are induced. Disturbances in soil microbial activity indirectly affect the enzymatic activity of the soil ecosystem. Suppression of the activity of soil enzyme may be due to increased mortality of microorganisms triggered by toxic doses of pesticides. The relevance among fluopyram, soil microbial communities and soil enzymes requires further research.

## Conclusion

The study suggested that chemigation system is beneficial not only for farming water usage in karst landscape in Guangxi, but also in controlling soil-disseminated disease efficaciously, stably and safely.

## Author Contributions

**Conceptualization:** Wenwei Tang.

**Methodology:** Jinzhao Li, Wenwei Tang.

**Validation:** Jinzhao Li, Cancan Wang, Haiou Lin.

**Writing – original draft:** Jinzhao Li.

**Writing – review & editing:** Saqib Hussain Bangash, Dongqiang Zeng, Wenwei Tang.

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
