## [Decision Letter · Decision Letter 0]

26 Mar 2020

PONE-D-20-00394

Efficacy of fluopyram applied by chemigation on controlling eggplant root-knot nematodes (Meloidogyne spp.)

PLOS ONE

Dear Dr. Tang,

Thank you for submitting your manuscript to PLOS ONE. After careful consideration, we feel that it has merit but does not fully meet PLOS ONE’s publication criteria as it currently stands. Therefore, we invite you to submit a revised version of the manuscript that addresses the points raised during the review process.

Especially, I request you to focus on the following areas:

The manuscript text needs extensive revision for English. The methodologies need more clarity in the descriptions. The Discussion needs to be more focused with the currently generated data as well as addition of already published and relevant references. The authors may also add additional data (if already generated) as suggested by one of the reviewers. Particularly, the data on remained nematode density in the soil on root knot disease parameters like disease incidence and disease severity compared to control will be of much help.We would appreciate receiving your revised manuscript by April 30, 2020. To enhance the reproducibility of your results, we recommend that if applicable you deposit your laboratory protocols in protocols.io, where a protocol can be assigned its own identifier (DOI) such that it can be cited independently in the future. For instructions see: http://journals.plos.org/plosone/s/submission-guidelines#loc-laboratory-protocolsPlease include the following items when submitting your revised manuscript:A rebuttal letter that responds to each point raised by the academic editor and reviewer(s). This letter should be uploaded as separate file and labeled 'Response to Reviewers'.A marked-up copy of your manuscript that highlights changes made to the original version. This file should be uploaded as separate file and labeled 'Revised Manuscript with Track Changes'.An unmarked version of your revised paper without tracked changes. This file should be uploaded as separate file and labeled 'Manuscript'.

We look forward to receiving your revised manuscript.

Kind regards,

Birinchi Sarma, PhD

Academic Editor

PLOS ONE

Journal Requirements:

Reviewers' comments:

Reviewer's Responses to Questions

**Comments to the Author**

1. Is the manuscript technically sound, and do the data support the conclusions?

Reviewer #1: Yes

Reviewer #2: Yes

Reviewer #3: Partly

Reviewer #4: Yes

2. Has the statistical analysis been performed appropriately and rigorously? 

Reviewer #1: Yes

Reviewer #2: I Don't Know

Reviewer #3: I Don't Know

Reviewer #4: I Don't Know

3. Have the authors made all data underlying the findings in their manuscript fully available?

Reviewer #1: Yes

Reviewer #2: Yes

Reviewer #3: Yes

Reviewer #4: Yes

4. Is the manuscript presented in an intelligible fashion and written in standard English?

Reviewer #1: Yes

Reviewer #2: No

Reviewer #3: No

Reviewer #4: Yes

5. Review Comments to the Author

Reviewer #1: please follow instructions to author. Introduction is sufficient . Results and discussion are sufficient. but the titles of figures are not found above each figure. please correct the mistakes and comments present in the text.

Reviewer #2: This manuscript is “Efficacy of fluopyram applied by chemigation on controlling eggplant root-knot nematodes (Meloidogyne spp.) and its effects on soil properties”.

The following are my comments and critique:

1. The manuscript title in text is different. “Efficacy of fluopyram applied by chemigation on controlling eggplant root-knot nematodes (Meloidogyne spp.) and its effects on soil properties” or “Efficacy of fluopyram applied by chemigation on controlling eggplant root-knot-nematodes (Meloidogyne spp.)”

2. The manuscript needs to be edited for grammar and syntax.

3. The introduction is generally relevant but unsufficient information about the previous study findings for readers. Previous studies on fluopyram to control RKN should be given in introduction.

4. The methods are generally appropriate.

5. The results are clear.

6. In discussion, presented data should be compared with previous findings. Some information is general concept. For example last paragraph in discussion section.

Reviewer #3: The manuscript gives some interesting information on a new nematicide and its effect on root-knot nematodes and soil microbial community (the latter probably should be included in the title). There are several issues that need to be addressed if this manuscript is to be accepted. The paper is difficult to read and the authors need to have this manuscript checked for English. There has been several recent publications on this nematicide, including on chemigation, and the authors should at least include some of this work in their references. The experimental design is not clearly explained - not sure how many replications are there and the sampling procedure is confusing - would be good to include a figure / diagram to clarify the sampling methodology. Also, the Baermann funnel method used to extract nematodes (need to give the correct (=original) reference for this method) is typically used to collect active nematodes from soil, not sure why the authors used a probe (which probe?) to verify if nematodes were dead or alive (if nematodes were dead, it's probably because they died from lack of oxygen in the funnel neck because they were not tapped off soon enough). Overall, the methodology lacks detail and is difficult to read and understand. The data tables and figures are also very busy and again don't help the readability of the paper. It also seems like this is just a single field experiment - duplicating the trial would strengthen the data and make this more relevant.

Reviewer #4: Actually I expected from the author to test remained nematode density in the soil on root knot disease parameters like disease incidence and disease severity compared with control, and also its effect on plant growth & yield parameters like fresh and dry weight of each of root and vegetative systems , plant hight, fruit weight and size ..etc, but unfortunately I didn't find such data. Also it was better if the author take more than one duration to determine Fluopyram residues in order to know the trend of product degradation exactly instead of consider one duration (30 DAT) for the purpose.

6. PLOS authors have the option to publish the peer review history of their article (what does this mean?). If published, this will include your full peer review and any attached files.

Reviewer #1: No

Reviewer #2: No

Reviewer #3: No

Reviewer #4: Yes: Dr. QAIS KADHIM ZEWAIN ALAZAWI

---

## [Author Response · Author response to Decision Letter 0]

20 May 2020

Thank you very much for your kind review and advice concerning our manuscript. The comments are valuable and helpful for revising and improving our paper and have provided good guidance for our studies. We have substantially revised our manuscript (PONE-D-20-00394) after reading the comments provided by the four reviewers. We employed an English-language editing service, Wiley Editing Service, to publish our wording. We also expanded part of the experiment, providing details in the current version. In discussion, we compared present research with previous findings.

---

## [Decision Letter · Decision Letter 1]

16 Jun 2020

Efficacy of fluopyram applied by chemigation on controlling eggplant root-knot nematodes (Meloidogyne spp.)and its effects on soil properties

PONE-D-20-00394R1

Dear Dr. Tang,

We’re pleased to inform you that your manuscript has been judged scientifically suitable for publication and will be formally accepted for publication once it meets all outstanding technical requirements.

Kind regards,

Birinchi Sarma, PhD

Academic Editor

PLOS ONE

Additional Editor Comments (optional):

Reviewers' comments:

Reviewer's Responses to Questions

**Comments to the Author**

1. If the authors have adequately addressed your comments raised in a previous round of review and you feel that this manuscript is now acceptable for publication, you may indicate that here to bypass the “Comments to the Author” section, enter your conflict of interest statement in the “Confidential to Editor” section, and submit your "Accept" recommendation.

Reviewer #1: (No Response)

Reviewer #2: All comments have been addressed

Reviewer #4: All comments have been addressed

2. Is the manuscript technically sound, and do the data support the conclusions?

Reviewer #1: Yes

Reviewer #2: Yes

Reviewer #4: Yes

3. Has the statistical analysis been performed appropriately and rigorously? 

Reviewer #1: Yes

Reviewer #2: I Don't Know

Reviewer #4: I Don't Know

4. Have the authors made all data underlying the findings in their manuscript fully available?

Reviewer #1: Yes

Reviewer #2: Yes

Reviewer #4: Yes

5. Is the manuscript presented in an intelligible fashion and written in standard English?

Reviewer #1: Yes

Reviewer #2: Yes

Reviewer #4: Yes

6. Review Comments to the Author

Reviewer #1: (No Response)

Reviewer #2: (No Response)

Reviewer #4: (No Response)

7. PLOS authors have the option to publish the peer review history of their article (what does this mean?). If published, this will include your full peer review and any attached files.

Reviewer #1: No

Reviewer #2: No

Reviewer #4: Yes: Prof.Dr. Qais K. Zewain

---

## [Editor Report · Acceptance letter]

23 Jun 2020

PONE-D-20-00394R1 

Efficacy of fluopyram applied by chemigation on controlling eggplant root-knot nematodes (Meloidogyne spp.)and its effects on soil properties 

Dear Dr. Tang:

I'm pleased to inform you that your manuscript has been deemed suitable for publication in PLOS ONE. Congratulations! Your manuscript is now with our production department. 

Kind regards, 

on behalf of

Dr. Birinchi Sarma 

Academic Editor

PLOS ONE